# INITIALIZING THE LAYER-WISE LEARNING RATE

## ABSTRACT

The standard method to assign learning rates has been to rely on the optimizer and to use a single, global learning rate across all its layers. We propose to assign individual learning rates as well, according to the layer-wise gradient magnitude at initialization. Even if individual layers are initialized to preserve gradient variance, architectural characteristics result in uneven gradient magnitude even when the network has not started training. We interpret this gradient magnitude as a measure of architecture-induced convergence bias, and adjust the layer-wise learning rate opposite to its gradient magnitude at initialization. This relative learning rate is maintained throughout the entire training scheme. Experiments on convolutional and transformer architectures on ImageNet-1k show improved accuracy and training stability.

## 1 INTRODUCTION

Appropriate hyperparameter selection remains a central component when training deep neural networks, and the network performance can be reliant on hyperparameter choices that is not apparent in the initial training stage. As such, it is customary to perform multiple full training runs over various hyperparameters when evaluating the performance of various techniques and architectures (Schmidt et al., 2021). Among them, perhaps the most important hyperparameter is the learning rate, which is multiplied at the final optimizer pipeline when calculating the step size.

Proposals to adjust the layer-wise learning rate can be traced back to 2002 (LeCun et al., 2002), and the consensus was that the need for different learning rates is automatically handled by adaptive optimization methods (Duchi et al., 2011; Kingma & Ba, 2015). Explicitly assigning layer-wise learning rates would mean updating the weights with different step sizes even if individual parameters were to have the same in-training time gradient or statistics, and the literature has been hesitant to suggest against the standard of using a single global learning rate when training from scratch.

Works that explicitly adjust the layer-wise learning rate were generally performed on transfer learning or finetuning settings (Donahue et al., 2014; Sharif Razavian et al., 2014; Kumar et al., 2022; Lee et al., 2023) in order to selectively train certain layers based on layer-wise characteristics such as feature generality (Azizpour et al., 2015; Cui et al., 2018; Michel et al., 2019). Such works mostly drop learning rates of specific layers to zero, effectively "freezing" them as further training of the pretrained layer is likely to worsen feature generality and performance.

However feature generality is already hard-coded in architectural characteristics such as layer depth, and differences in layer-wise convergence are observable during training (Zeiler & Fergus, 2014; Yosinski et al., 2014; Chen et al., 2023b), so a naturally following argument is that adjusting the layer-wise learning rate could be applicable outside of finetuning settings. The difficulty would be in determining the precise learning rate values that strike a careful balance between inter-layer overfitting and under-fitting, as the network has not been pre-trained on separate tasks and has to be trained from scratch. Previous works that assigned more granular layer-wise learning rates mainly followed the vague consensus it is depth dependent, resulting in heuristics that adjust the learning rate from low to high according to depth (Brock et al., 2017; Ro & Choi, 2021; Yang et al., 2019).

This work started from an observation that adjusting the layer-wise learning rate can potentially improve training on settings where it is not explained and reproducible by adaptive methods. It motivated a search for a concrete statistical value that represents the layer-wise characteristics, and thus is usable as basis for explicitly adjusting the relative layer-wise learning-rates.

On the other hand, various works have shown that the training dynamics is both very different and important at initialization (Achille et al., 2019; Frankle et al., 2020; Jastrzebski et al., 2020; Frankle & Carbin, 2019), and we observed that the gradient magnitude at initialization seems to correlate with the layer-wise characteristics with respect to depth. That is, deeper layers that are generally considered to converge slower had lower gradient magnitude at initialization compared to shallower layers, which are widely considered to converge earlier.

We develop a scheme that adjusts the layer-wise learning rate opposite to the gradient magnitude at initialization. The gradient is measured when layer weights are initialized to preserve their backward gradient variances, so differences in gradient magnitude are induced by architectural factors such as layer connectivity, use of activation functions and normalization layers, which by normalizing the forward activations also modifies the backward gradient magnitude. The proposed method considers the uneven gradient magnitude a naturally occurring consequence due to architecture-induced convergence bias and adjusts the learning rate accordingly, which is in contrast to methods that explicitly normalize the gradient magnitude (Yu et al., 2017; Zhang et al., 2018b; 2020a).

Experiments on ImageNet-1k classification show improved performance and training stability for ResNet-50 (He et al., 2016) and ViT-S/16 (Dosovitskiy et al., 2021), an intriguing phenomenon given the simplicity of the method as it does not require prior knowledge of architecture specific training dynamics or complicated theoretical analysis (Dong et al., 2021; Poole et al., 2016; Schoenholz et al., 2017). Inspecting the actual layer-wise learning rates values show that low learning rates are assigned to low level layers and high learning rates to deeper, high level layers, but with additional subtleties and details as they are assigned per-layer granularity. More importantly, it suggests that adjusting the learning rate is an appropriate solution to handle the diverse gradients that occur not only at initialization but also during prolonged training.

## 2 RELATED WORKS

Weight initialization schemes were initially proposed as a solution to the exploding/vanishing activation/gradient problem in vanilla feed-forward networks (Glorot & Bengio, 2010; He et al., 2015). However, preserving both the forward activation and backward gradient variance requires specialized initialization schemes such as orthogonal initialization for layers with different input and output neuron number (Mishkin & Matas, 2016; Pennington et al., 2017; Xiao et al., 2018). Batch normalization and layer normalization are widely used to train deep networks (Ioffe & Szegedy, 2015; Ba et al., 2016; Santurkar et al., 2018), and subsequent works explored the interaction between initialization and normalization to more accurately isolate the cause of training instability (Balduzzi et al., 2017; Hanin & Rolnick, 2018; Zhang et al., 2019b; De & Smith, 2020; Daneshmand et al., 2020). Recent works extend such analysis to the more recent transformer networks (Xiong et al., 2020; Liu et al., 2020; Huang et al., 2020; Zhang et al., 2019a), while others strived for architecture agnostic initialization schemes (Dauphin & Schoenholz, 2019; Zhu et al., 2021).

Commonly used optimizers such as SGD and Adam (Kingma & Ba, 2015) adjust the step size according to the gradient size and variance history, and many works that advocated the need for layer-wise learning rate adjustment are based on layer-wise gradient normalization and weight norm scaling (Singh et al., 2015; You et al., 2017; Yu et al., 2017; You et al., 2020; Xiao et al., 2019; Ginsburg et al., 2019). Another direction previous works indirectly modified the learning rate is through the use of scale factors, most commonly included at the residual branches (Karras et al., 2018; Hayou et al., 2021; Liu et al., 2020; Touvron et al., 2021; Noci et al., 2022). Such scale factors can affect both the gradient size and effective weight deviation of the corresponding layer, resulting in different effects depending on whether the optimizer decides the step size based on gradient magnitude or variance (Balles & Hennig, 2018).

Many works can be interpreted as adaptively assigning layer-wise learning rates, such as adaptive methods (Duchi et al., 2011), efficient approximation of the Hessian in second order optimizers (Martens & Grosse, 2015; Gupta et al., 2018; Yao et al., 2021) and the use of hypergradient for automatic hyperparameter adjustment in layer-wise meta-learning settings (Antoniou et al., 2019; Baik et al., 2020; Tang et al., 2021; Chen et al., 2022). Other approaches include the use of mutual information (Vasudevan, 2018), reinforced learning (Mahmud et al., 2022), expected parameter scale size (Tian & Parikh, 2022), and in context of disentangling step size from learning rate schedules (Agarwal et al., 2020) and zero-shot hyperparameter transfer (Yang et al., 2022).

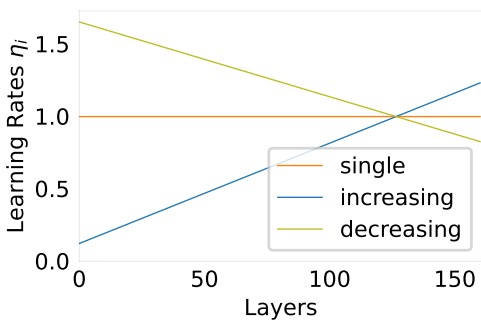 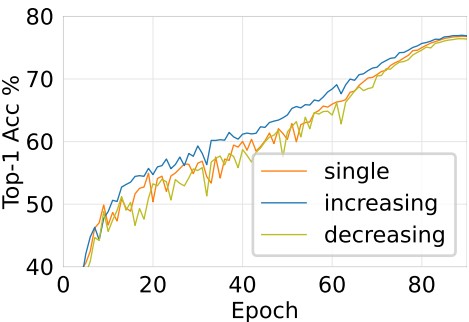

Figure 1: **Left**: Naïve learning rate increasing/decreasing scheme according to depth on ResNet-50. **Right**: Validation top-1 accuracy. The single, increasing, decreasing learning rate schemes achieve 76.80%, 76.91%, 76.39% final accuracy.

## 3 LAYER-WISE LEARNING RATES

A typical neural network training revolves around optimizing a training loss that represents the network performance on the train data. Let $L(X; \boldsymbol{\theta}) = \frac{1}{|X|} \sum_{x \in X} l(x; \boldsymbol{\theta})$ be the average loss of the model with weights $\boldsymbol{\theta}$ on a minibatch of samples $X$, where $\boldsymbol{\theta} = \{\boldsymbol{W}_1, \boldsymbol{W}_2, ..., \boldsymbol{W}_M\}$ is the set of layer weights for the scale, bias, convolution and linear layers. The gradient $\boldsymbol{g}_{X,\boldsymbol{\theta}} = \nabla_{\boldsymbol{\theta}} L(X; \boldsymbol{\theta})$ is obtained through back-propagation and is fed as input to the optimizer $O(\boldsymbol{g}_{X,\boldsymbol{\theta}}, \gamma)$. Internally, the optimizer decides the step sizes depending on the tracked historical gradient, and then multiplies it by the global learning rate $\gamma$ to output the actual step sizes used to update the layer weights as below, where we extracted the learning rate multiplication out of the optimizer.

$$\Delta\boldsymbol{\theta} = \gamma O'(\boldsymbol{g}_{X,\boldsymbol{\theta}}) \tag{1}$$

During training from scratch the global learning $\gamma$ is optionally warmed up from 0 to the initial learning rate, a hyperparameter typically set beforehand, and then decayed to 0 through schedules such as cosine or step-wise decay. In this paper we refer to layer-wise learning rates $\eta_i$ as additionally multiplying the step sizes with different relative values per layer, as represented below.

$$\Delta\boldsymbol{W}_i = \gamma\eta_i O'(\boldsymbol{g}_{X,\boldsymbol{W}_i}) \tag{2}$$

Recently Lee et al. (2023) showed the type of distribution shift can influence which layers should be retrained in finetuning, and several works investigated the effects of regularizing effects of high or low global learning rate (Goyal et al., 2017; Li et al., 2019), but the effects of layer-wise learning rate remain unclear in general settings. A simple interpretation is that a higher learning rate will lead to a stronger optimization focus on the respective layers, similar to how it is used on finetuning and transfer learning. On the other hand, increased stochastic noise due to higher initial learning rates has been linked with stronger regularization and improved generalization, so conceivably increasing the learning rate of certain layers will train the network to be more resilient to their weight deviations.

Figure 1 shows the top-1 validation accuracy on ImageNet-1k when the layer-wise learning rate is adjusted in increasing/decreasing order according to depth on ResNet-50 under typical settings. We would like to note that this is a setting where adaptive methods are known to achieve lower validation accuracy despite their improved convergence. For demonstration purposes the relative learning rate is increased from 0.1 to 1 or decreased from 2 to 1, and then normalized so that the average per-parameter learning rate is 1. A naïve learning rate increasing scheme results in a 0.11% final validation accuracy improvement compared to a single learning rate scheme, and the decreasing scheme results in a 0.41% lower final validation accuracy.

While this simple experiment suggest a learning rate increasing scheme can have benefits, there are many caveats that question whether such a scheme is practical. There are little theoretical justifications for performing different step sizes even if the gradient history is identical, and the motivation for layer-wise learning rate mainly stems from a post observation that convergence speed is dependent of depth, making it infeasible to apply in practice on diverse architectures. Perhaps due to the small performance gains, many works instead focused on faster wall clock training time from the reduced computation overhead when freezing weights (Xiao et al., 2019; Bragagnolo et al., 2022).

## 3.1 REGULARIZING ARCHITECTURE-INDUCED CONVERGENCE BIAS

---

**Algorithm 1:** Layer-wise learning rate initialization of neural networks

---

    **Input:** Model weights $\boldsymbol{\theta} = \{\boldsymbol{W}_1, ..., \boldsymbol{W}_M\}$ and numbers of parameters $\{m_1, ..., m_M\}$
    **Output:** Relative learning rates $\eta_i$

1: $G_{1...M} \leftarrow 0$
2: **For** $\boldsymbol{W}_i \in \boldsymbol{\theta}$
3:     Initialize conv/linear layers from $N(0, \sqrt{1/f_{out}})$, scale layers to **1**, bias layers to **0**
4: **For** $t \leftarrow 1$ **to** $T$
5:     Sample minibatch $X_t$ from training set
6:     $\boldsymbol{g}_{X_t,\boldsymbol{\theta}} \leftarrow \nabla_{\boldsymbol{\theta}} L(X_t; \boldsymbol{\theta})$
7:     **For** $i \leftarrow 1$ **to** $M$
8:         $G_i \leftarrow G_i + \frac{1}{m_i} \sum \|\boldsymbol{g}_{X_t,\boldsymbol{W}_i}\|_1$
9: **For** $i \leftarrow 1$ **to** $M$
10:     $\tilde{\eta}_i \leftarrow \frac{1}{\sqrt{G_i}}$
11: $m_{sum} = \sum_{i=1}^{M} m_i, \tilde{\eta}_{sum} = \frac{1}{m_{sum}} \sum_{i=1}^{M} m_i \times \tilde{\eta}_i$
12: **For** $i \leftarrow 1$ **to** $M$
13:     $\eta_i \leftarrow \frac{\tilde{\eta}_i}{\tilde{\eta}_{sum}}$

---

Previous works that adaptively modified the learning rate of weights mostly did so according to convergence, measured by tracking changes in gradient, weight, neuron output or loss sensitivity (Raghu et al., 2017; Xiao et al., 2019; Bragagnolo et al., 2022; Liang et al., 2022; Du et al., 2022). However it is difficult to determine before training has ended if the current weight is close to the final value, and weights considered to have converged may require further training as the remaining weights are modified (Bragagnolo et al., 2022). Preemptive convergence of selected weights may also make the network brittle to weight deviations of those frozen prematurely, resulting in reduced generalization. A recurring theme is the reliance on in-training time statistics, which has been explored extensively in various optimizer literature and may not appropriately incorporate more global aspects of the training dynamics (Micaelli & Storkey, 2021). As an alternative method, we propose to assign layer-wise learning rates opposite to the gradient magnitude at initialization.

The training objective of the model is to minimize the loss, and assuming a smooth, convex surface a minima is likely to have small to zero gradient around it. An alternative formulation of the training objective becomes minimizing the gradient, which for individual layers becomes minimizing their output activation gradient. Even if individual linear and convolution layers are initialized to preserve backward gradient variance, architectural characteristics such as non-linear activation functions and normalization layers modify the backward gradient as it is back-propagated through the model, and hence result in uneven gradient magnitude across the layer weights. The proposed algorithm sets the layer-wise learning rate to regularize the convergence bias, measured by the gradient magnitude of the weights at initialization, and we empirically found it is better to scale the learning rate by the square root of gradient magnitude.

The layer-wise learning rate initialization is outlined in Algorithm 1. It starts by initializing all convolution/linear weights from random numbers sampled from $N(0, \sqrt{1/f_{out}})$, where $f_{out}$ is the fan out number. It is a basic initialization scheme that preserves the backward gradient variances if the weights and gradients are i.i.d. (Glorot & Bengio, 2010). As standard, scale layers are initialized to 1 and bias layers to 0. The class token, position embedding layer and relative position bias layer in transformer architectures are treated as bias layers and initialized to 0. Next we collect the layer-wise gradient magnitude for $T$ iterations while the model weights are fixed at initialization. The layer-wise gradient magnitude is collected per parameter as the learning rate is applied on a per parameter basis. The relative layer-wise learning rates are adjusted inversely proportional to the square root of the gradient magnitude, and normalized to unit parameter-wise learning rate.

Table 1: Final top-1 validation accuracy for ImageNet-1k classification. Trained without gradient clipping and label smoothing.

| Model | #Params | Optimizer | Data Augmentation | Epochs | Learning Rate | |
|---|---|---|---|---|---|---|
| | | | | | Single | Layer-wise |
| ResNet-50 | 25.56M | SGD | basic | 90 | 76.80 | **77.12(+0.32)** |
| | | | basic | 200 | 77.11 | **77.98(+0.87)** |
| | | | strong | 300 | 78.41 | **79.12(+0.71)** |
| | | AdamW | basic | 90 | 76.40 | **76.67(+0.27)** |
| | | | basic | 200 | 76.44 | **77.12(+0.68)** |
| | | | strong | 300 | 78.49 | **78.96(+0.47)** |
| ConvNeXt-T | 28.59M | AdamW | strong | 300 | 80.49 | **80.56(+0.07)** |
| ViT-S/16 | 22.05M | SGD | basic | 300 | 65.53 | **70.29(+4.76)** |
| | | AdamW | basic | 300 | 74.11 | **74.58(+0.47)** |
| | | | strong | 300 | 78.00 | **78.25(+0.25)** |
| Swin-T | 28.29M | AdamW | strong | 300 | 79.84 | **79.96(+0.12)** |

Table 2: Comparison with layer-wise gradient normalization techniques when training ResNet-50 on ImageNet-1k for 90 epochs

| SGD | SGD-LW | SGD-NG | Adam-NG | LARS | LAMB |
|---|---|---|---|---|---|
| 76.80 | **77.12** | 76.85 | 76.33 | 76.71 | 76.43 |

## 4 EXPERIMENTS

We evaluate the performance gain of the proposed layer-wise learning rate scheme on convolutional networks and vision transformers for ImageNet-1k classification when trained from scratch under basic Inception-style preprocessing and strong data augmentation. We find it improves training on both convolution networks and vision transformers even though architecture specific training dynamics were not explicitly utilized to decide the layer-wise learning rates. We also inspect the assigned layer-wise learning rates, which in general increase according to depth, but also show other intricacies.

For hyperparameters such as initial learning rate, effective batch size and stochastic depth we mostly rely on the values reported in Chen et al. (2023a) for resnets and vision transformers. For other architectures we rely on the author reported values to minimize the cost of hyperparameter search. When training ResNet-50 with SGD, we use a effective batch size of 256 and weight decay of 1e-4 similar to Goyal et al. (2017). We report the exact hyperparameters used in the Appendix. All experiments were trained using a cosine decay learning rate schedule with warmup, image resolution of 224 × 224 and default momentum/beta hyperparameters for SGD and AdamW (Loshchilov & Hutter, 2019). When adjusting the layer-wise learning rates we collect the gradient for 1 epoch, which takes 0.3˜1.1% percent of the total training time depending on the total training epochs.

The experiments were performed using pytorch (Paszke et al., 2019) and timm (Wightman, 2019) for transformer model architectures and data preprocessing. Each experiment were run on environments consisting of a single GPU with 24GB of DRAM, which fits a batch size of 256 for all reported experiments, and perform gradient accumulation to simulate larger effective batch sizes. We do not perform gradient clipping or label smoothing, but include strong data augmentations to demonstrate the effectiveness of appropriate layer-wise learning rate under more difficult training settings. For strong data augmentation we use a combination of RandAugment with layer 2 and magnitude 10 (Cubuk et al., 2020) and Mixup with strength 0.5 (Zhang et al., 2018a). All reported ImageNet-1k

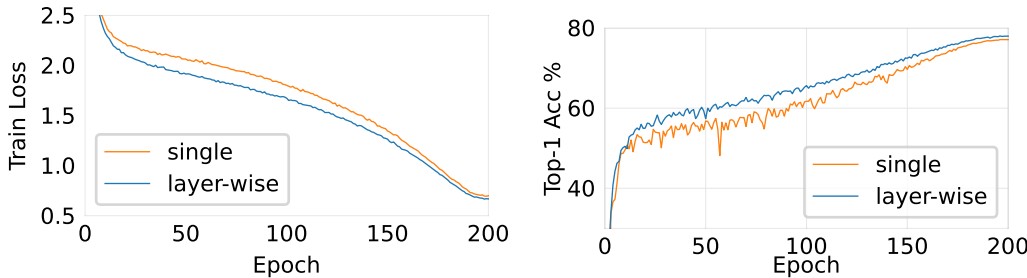

Figure 2: ResNet-50 train loss and validation accuracy with basic data augmentation and SGD.

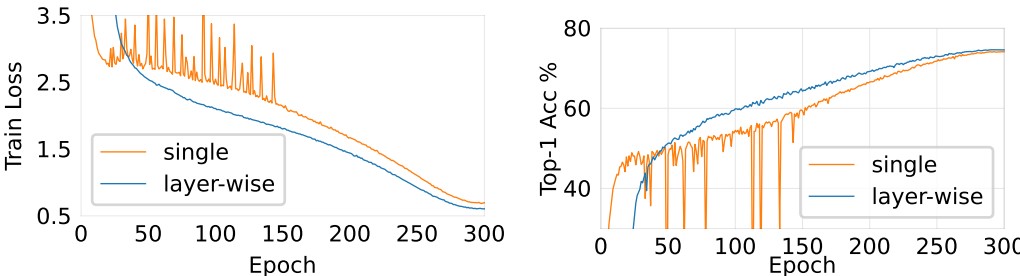

Figure 3: ViT-S/16 train loss and validation accuracy with basic data augmentation and AdamW.

experiments took in total ∼115 GPU days of training and we report the final top-1 accuracy from a single training run in Table 1.

## 4.1 IMAGENET-1K RESULTS

It it well known that SGD has strong performance when training convolutional networks under simple data preprocessing pipelines, and we can see under basic Inception-style augmentation SGD has a higher top-1 validation accuracy than Adam for ResNet-50. While a naïve increasing learning rate leads to a 0.10% accuracy increase as shown in Figure 1, the proposed layer-wise learning rate increases the accuracy by a more tangible 0.32%. The train loss and accuracy curves in Figure 2 show that the proposed layer-wise learning rate achieves a lower final train loss, suggesting that the benefit arise from an improved fitting of the training dataset. Another notable benefit is a larger accuracy increase for ResNet-50 when trained for a longer 200 epochs with layer-wise learning rates. When trained for 200 epochs, the accuracy increase with a single learning rate is 0.31%, but when trained with layer-wise learning rates the accuracy increases by a much larger 0.86%. We would like to note that such performance gain is contrary to the common belief that 90 epochs is sufficient to train ImageNet-1k on ResNet-50 under simple settings. Improved accuracy when the layer-wise learning rate is adjusted can be also observed under strong data augmentations and when using AdamW, showing that the benefits of layer-wise learning rate is observable regardless of the optimizer choice and under harder training difficulty. Overall it suggests that there has been underutilized capacity in resnets, which is difficult to leverage with a single learning rate.

Training vision transformers is known to be reliant on adaptive optimizers (Zhang et al., 2020b), and although adjusting the layer-wise learning rate drastically improves performance when trained with SGD, it does not make it as performant as AdamW. Similar to convolutional networks we can see the proposed layer-wise learning rate scheme improves final validation accuracy regardless of the data augmentation strength, but comparatively less than the improvement in ResNet-50. Perhaps more impressive is the improved stability visible in the train loss curves in Figure 3. Whereas a single learning rate shows high fluctuations in both the train loss and validation accuracy in the initial stage, assigning layer-wise learning rates removes the fluctuations and result in markedly lower train loss, showing that the network is trained to better fit the training dataset. Interestingly, the train loss improvement is visibly larger in vision transformer compared to resnet, but it does not result in a comparable accuracy improvement. The final validation accuracy improvement for ViT-S/16 is a lower 0.25% compared to the 0.47% for ResNet-50 when trained with AdamW under strong

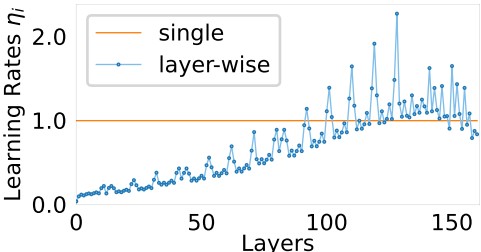 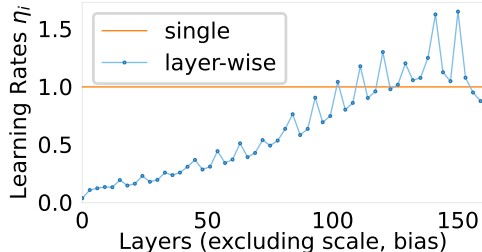

Figure 4: **Left**: Assigned learning rates to ResNet-50 when trained for 200 epochs using SGD. **Right**: Same as left but only showing convolution/linear layers, which comprise 99.8% of total parameter count.

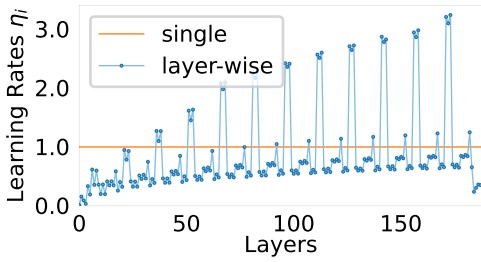 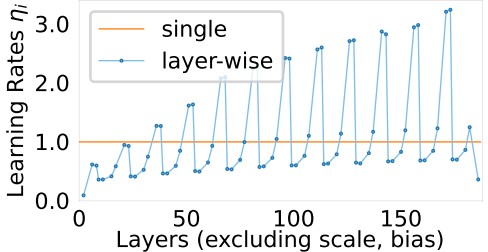

Figure 5: **Left**: Assigned learning rates to ViT-S/16 when trained for 300 epochs with basic augmentation using AdamW. **Right**: Same as left but only showing convolution/linear layers, which comprise 99.4% of total parameter count.

data augmentation, suggesting there is a overfitting problem that is known to arise when training transformer architectures on image data. While the validation performance increase is not very significant, the large improvement when trained with SGD and the visibly improved train loss and stability show that appropriate layer-wise learning rate is also beneficial to transformer architectures.

## 4.2 INSPECTING ASSIGNED LEARNING RATES

Most network architectures consist of multiple repeated blocks of specific layer configurations, with a separate stem that processes the input RGB image and a fully connected layer at the end that outputs the final logits. ViT-S/16 has 12 attention blocks while ResNet-50 has a 3-4-6-3 convolution block configuration. The assigned learning rates similarly show repeated patterns that reflects the regular block structure of the architecture, which is especially pronounced in vision transformer architectures. ResNet-50 show a stronger tendency to increase with respect to depth, although interestingly there is a slight learning rate decay near the final layer, starting at the stage consisting of 3 blocks that precedes the final layer.

Figure 4 shows the assigned layer-wise learning rates of a training run for ResNet-50. Since there are numerous layers in a typical network, especially when considering scale and bias as separate layers, we separately show the assigned learning rates for the convolution and linear layers, as they comprise most of the model parameter number. ResNet-50 consist of multiple residual bottleneck blocks which individually is made up of 3 convolution layers, and the assigned learning rates reflect the repetition of bottleneck blocks though the pattern of high learning rate for the first convolution layer and lower learning rates to the subsequent two convolution layers. The first convolution layer of a bottleneck block has a lower gradient magnitude due to the combined effect of increasing input activation magnitude as more residual connections are added and the normalization effect of the following batchnorm layer. In order to normalize the large activation variance, the directly following batchnorm layer can be thought as multiplying it by a low scale factor, which in the backward pass reduces the gradient magnitude to the preceding convolution layer. It also leads to the low gradient magnitude of the batchnorm scale and bias layer in the third convolution layer in the preceding residual block, and the unusually high learning rates assigned to the bias layer are visible as spikes in Figure 4. We note a more formal analysis for the relationship between gradient variance and

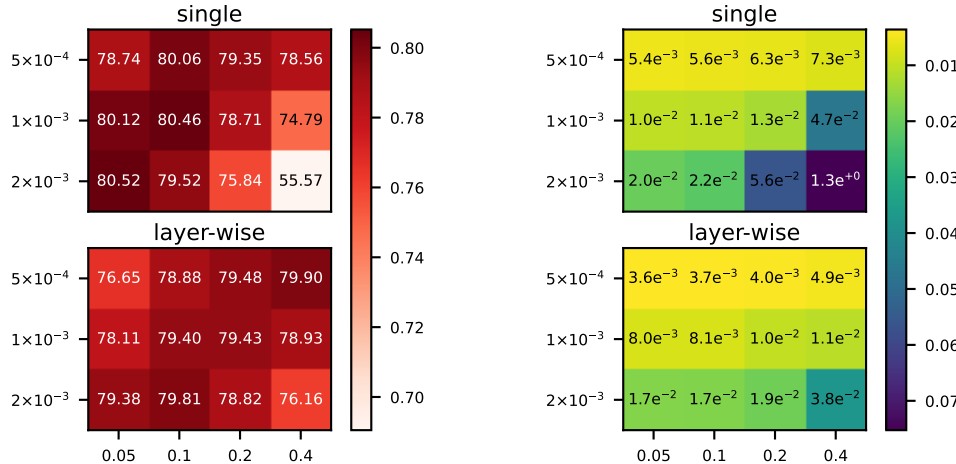

Figure 6: Final top-1 accuracy (**left**) and train loss (**right**) for ResNet-50 on CIFAR-100 under different initial learning rates (*x* axis) and weight decay values (*y* axis). Each result is averaged over 5 experiments.

batch normalization is given in Zhu et al. (2021). Another notable observation is that the first 7 by 7 stem convolution layer is assigned a low learning rate of 0.038, which is considerably lower than the 0.098 assigned to the directly succeeding batchnorm scale layer.

A vision transformer block is comprised of a self attention block followed by a multilayer perceptron block. The self attention block extracts the query, key and value from the input, performs self attention and then passes the result into a projection layer before summing it with the input in a residual manner. The spiking learning rates in Figure 5 correspond to the query and key layers, which show that their gradient are very low compared to other layers at initialization. Similar findings have been reported in Noci et al. (2022), where it only analyzed the gradient discrepancy in the attention block and proposes to introduce an inverse temperature scaling. Our approach adjusts the learning rate and is applied equally to all layers in a network. The succeeding value and projection layers are assigned comparatively lower learning rates, slightly lower than the two succeeding linear layers that comprise the multilayer perceptron block. The first convolution layer in the stem patch embedding is assigned a low learning rate of 0.092, while the class token layer and the position embedding layer, which represented as the first two layers in Figure 5, are assigned a very low 0.012 and 0.16. We provide figures for ConvNeXt-T and Swin-T in the appendix.

### 4.3 CIFAR-100 RESULTS

We also evaluate the effect of layer-wise learning rates on CIFAR-100 classification when trained from scratch on ResNet-50 using SGD. We train for 200 epochs with 1 epoch warmup under the standard data preprocessing of random cropping with padding 4 and horizontal flipping. We use a batch size of 256 and perform cosine learning rate decay.

The single learning rate achieves a top-1 accuracy of 80.52% when trained with a learning rate of 0.05 and weight decay of 0.002, while under the same hyperparameters layer-wise learning rate achieves a significantly lower 79.38%. Investigating the train loss shows that although layer-wise learning rate results in a lower final train loss of 0.017 compared to 0.020 of single learning rate, it does not result in improved final test accuracy. While the accuracy of layer-wise learning rate improves up to 79.90% when increasing the initial learning rate to 0.4, it is not an improvement over the best accuracy from the single learning rate. We suspect the smaller dataset size and difficulty means the accuracy is more influenced by other factors, such as the regularizing effect of high learning rates discussed in Section 3. In fact, one could even argue that the architecture itself has been designed to minimize the generalization gap when trained with a single learning rate. Even if the layer-wise learning rate does not improve the top-1 accuracy, the improved convergence of appropriate layer-wise learning is shown in that it can handle larger learning rates and weight decay values.

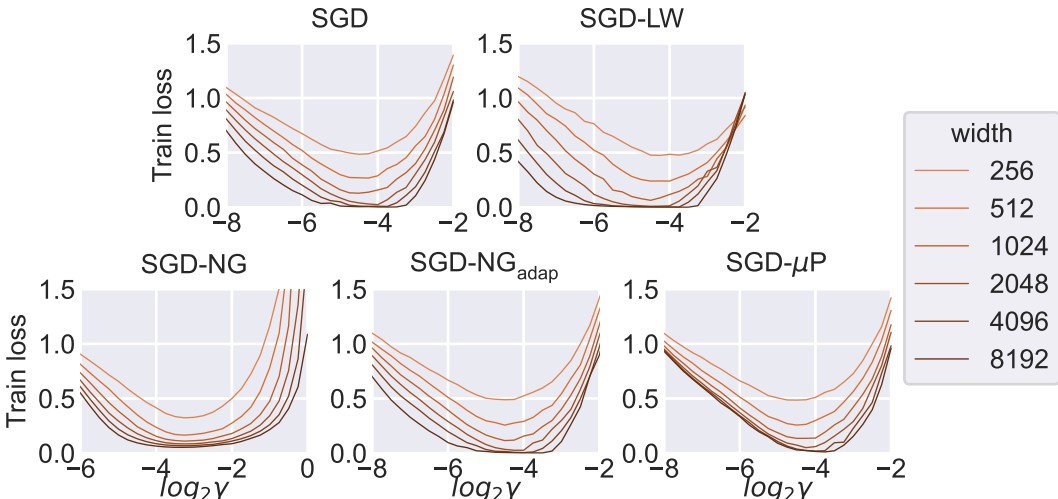

Figure 7: Train loss for 2-layer MLP trained on CIFAR-10. From top left to right: (a) Standard SGD, (b) layer-wise learning rate initialization, (c) per-layer normalized gradient, (d) per-layer normalized gradient and weight norm scaling and (e) maximal update parameterization ($\mu$P)

### 4.4 COMPARISON WITH RELATED WORK

A variety of optimizers such as SGD-NG (Yu et al., 2017) adaptively adjusts layer-wise learning rate by normalizing the layer-wise gradient norm or step size (You et al., 2020), and a recent work assigns layer-wise learning rates in context of zero-shot hyperparameter transfer between small and large models (Yang et al., 2022). Figure 7 shows the training performance when deliberately overfitting a 2-layer MLP of various sizes and learning rates for 20 epochs on CIFAR-10. We observe that simply replacing the gradient with layer-wise normalized gradient (SGD-NG) makes it mores stable over larger learning rates, but due to the loss of gradient norm information it is unable to reach low training loss despite the large model sizes. While additionally including weight norm based scaling (SGD-NG_adap) allows it to reach low training loss, the proposed layer-wise learning rate initialization is stable across wider learning rate values, showing the improved convergence available when appropriately utilizing inter-layer gradient norm information.

## 5 CONCLUSION

We propose a systematic layer-wise learning rate adjusting scheme that accounts for architecture-induced convergence bias and show that it can result in improved training performance and stability on convolutional and transformer architectures. The gradient magnitude at initialization is used as a measure of layer-wise convergence and the learning rate is adjusted to counteract such convergence bias. The gradient is measured when weights are initialized so that individual layers preserve the gradient variances and differences in gradient magnitude arise due to architectural characteristics. It suggests that adjusting the learning rate is an effective solution to handle the diverse gradients that arise in prolonged training of complex architectures, and that architectural characteristics such as layer-wise convergence can be inferred from statistics at initialization.

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

# A APPENDIX

Table 3: Deviation in per-layer assigned learning rates and accuracy for different T when compared to T=5004 (one epoch) on ResNet-50 with SGD for 90 epochs. The results are from identical weight initialization.

| T | max deviation % | average deviation % | validation accuracy % |
|---|---|---|---|
| 1 | 11.02 | 2.66 | 77.07 |
| 10 | 3.51 | 0.69 | 77.01 |
| 100 | 1.01 | 0.18 | 77.04 |
| 1000 | 0.32 | 0.06 | 77.07 |
| 5004 | 0 | 0 | 77.15 |

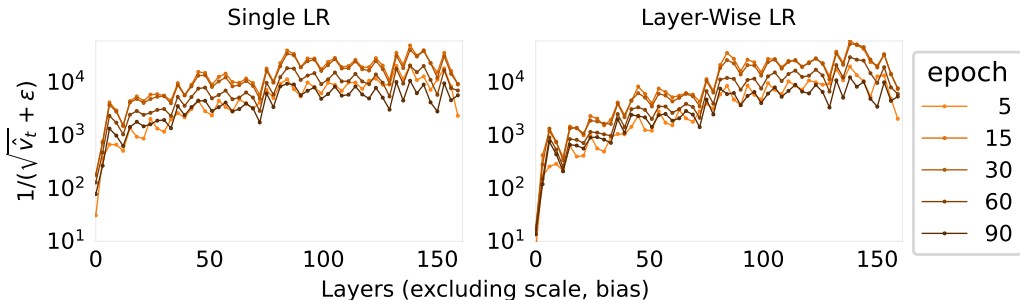

Figure 8: Average adaptive step size multiplier by second moment on AdamW when training ResNet-50 for 90 epochs. It remains mostly unchanged even with additional layer-wise learning rates. **Left**: Single learning rate, **Right**: Layer-wise learning rate.

Table 4: ImageNet-1k real, v2 performance improvement due to layer-wise learning rate.

| Model | #Params | Optimizer | Data Augmentation | Epochs | Accuracy ReaL | Accuracy V2 |
|---|---|---|---|---|---|---|
| ResNet-50 | 25.56M | SGD | basic | 90 | 83.51(+0.02) | 72.46(+0.51) |
| | | | basic | 200 | 83.97(+0.71) | 73.43(+0.65) |
| | | | strong | 300 | 85.43(+0.27) | 75.18(+1.20) |
| | | AdamW | basic | 90 | 82.93(+0.21) | 71.52(-0.15) |
| | | | basic | 200 | 83.12(+0.63) | 72.05(+0.37) |
| | | | strong | 300 | 85.24(+0.24) | 74.56(+0.26) |
| ConvNeXt-T | 28.59M | AdamW | strong | 300 | 85.57(+0.05) | 76.03(+0.23) |
| ViT-S/16 | 22.05M | SGD | basic | 300 | 77.07(+4.90) | 65.06(+5.41) |
| | | AdamW | basic | 300 | 80.45(+0.48) | 68.76(-0.25) |
| | | | strong | 300 | 84.13(+0.26) | 73.19(-0.13) |
| Swin-T | 28.29M | AdamW | strong | 300 | 85.11(-0.02) | 75.06(+0.10) |

For ConvNeXt-T low learning rates are assigned to the first convolution layer of the block while the subsequent two linear layers that comprise the remaining of the block are assigned larger learning rates. The learning rates of Swin-T shows a similar pattern to ViT-S/16, except that the relative posi-

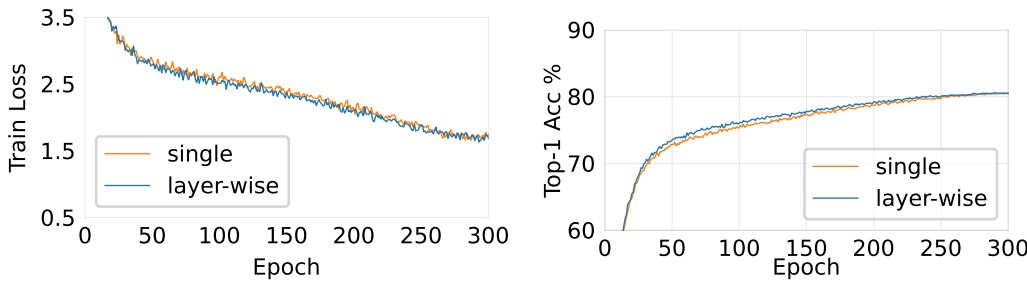

Figure 9: ConvNeXt-T train loss and validation accuracy with strong data augmentation.

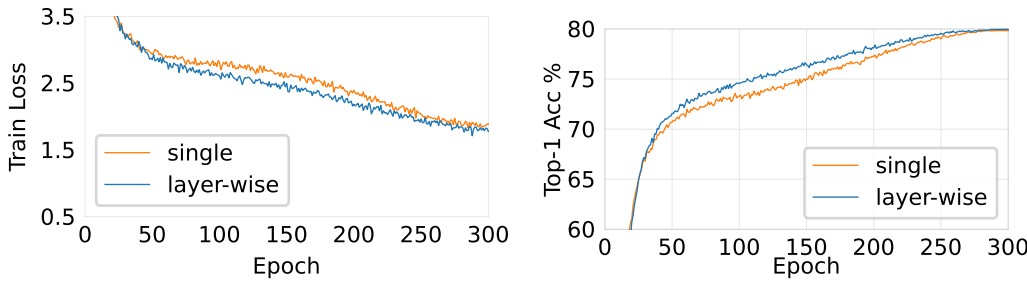

Figure 10: Swin-T train loss and validation accuracy with strong data augmentation.

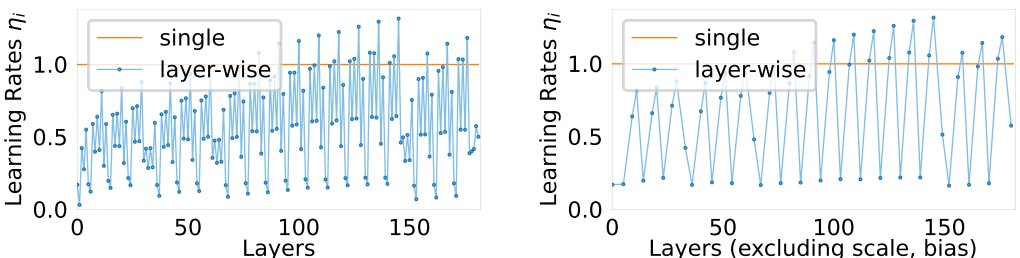

Figure 11: **Left**: Assigned learning rates to ConvNeXt-T when trained for 300 epochs with strong augmentation. **Right**: Same as left but only showing convolution/linear layers, which comprise 99.8% of total parameter count.

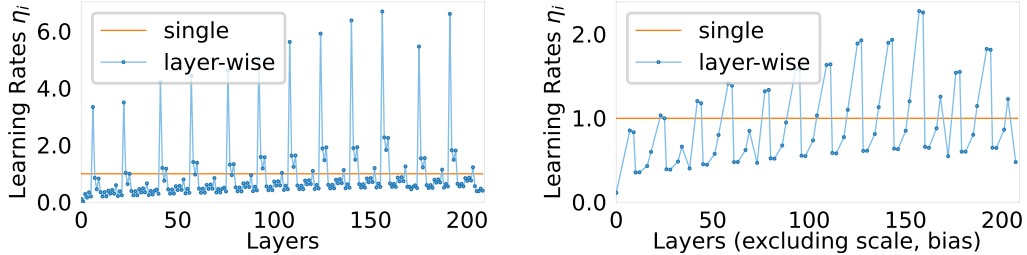

Figure 12: **Left**: Assigned learning rates to Swin-T when trained for 300 epochs with strong augmentation. **Right**: Same as left but only showing convolution/linear layers, which comprise 99.7% of total parameter count.

tion layers are assigned noticeably large learning rates. Both architectures show a weaker tendency of the assigned learning rates to increase with respect to depth compared to ResNet-50.

The SGD-NG and Adam-NG in Table 2 was trained with the same hyperparameters as in Table 1. For LARS we used the clipped variant LARC with learning rate of 0.2 and trust coefficient of 0.001 and for LAMB we used a batch size of 2k and 1.25 warmup epochs, polynomially decaying learning

Table 5: Experiment hyperparameters

| Model | Dropout | Stoch Depth | Data Augmentation | Optimizer | Batch Size | lr | wd |
|---|---|---|---|---|---|---|---|
| ResNet-50 | - | - | basic | SGD | 256 | 0.1 | 1e-4 |
| | - | - | strong | SGD | 256 | 0.1 | 1e-4 |
| | - | - | basic | AdamW | 1024 | 3e-3 | 0.1 |
| | - | - | strong | AdamW | 1024 | 3e-3 | 0.1 |
| ConvNeXt-T | - | 0.1 | strong | AdamW | 4096 | 4e-3 | 5e-2 |
| ViT-S/16 | 0.1 | 0.1 | basic | SGD | 4096 | 0.5/1.6 | 1e-4 |
| | 0.1 | 0.1 | basic | AdamW | 4096 | 1e-2 | 0.1 |
| | - | - | strong | AdamW | 4096 | 1e-2 | 0.1 |
| Swin-T | - | 0.2 | strong | AdamW | 1024 | 1e-3 | 5e-2 |

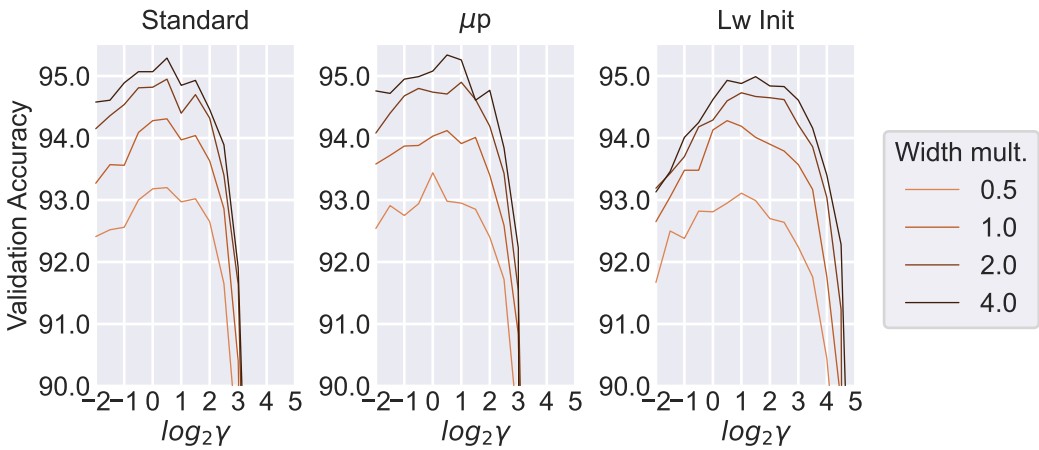

Figure 13: DavidNet with output multiplier of 0.125 on CIFAR-10 across different width and learning rates trained with SGD. **Left**: Default initialization and single learning rate, **Middle**: Maximal update parametrization ($\mu$P), **Right**: Layer-wise learning rate initialization.

rate schedule, weight decay of 1.5 applied only to the linear and convolution layers. The trust ratio multiplication was performed only to the linear and convolution layers.

Davidnet is a resnet variant designed for fast training on CIFAR-10, and the results in Figure 13 show similar trends to the CIFAR-100 results in Figure 6. The proposed layer-wise learning rate

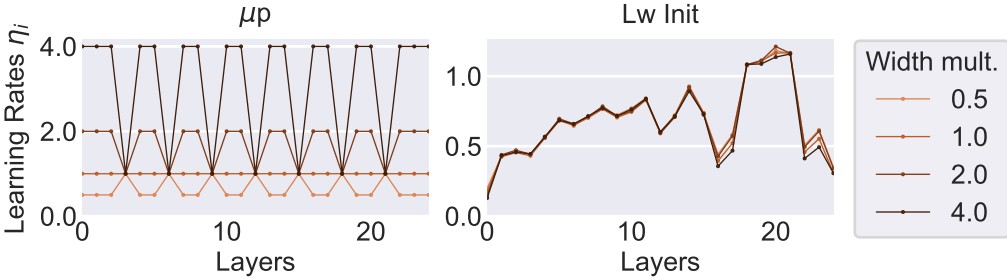

Figure 14: Assigned learning rates for DavidNet. **Left**: Maximal update parametrization ($\mu$P), **Right**: Layer-wise learning rate initialization.

initialization can handle larger learning rates, but the validation accuracy is slightly lower compared to plain SGD and maximal update parametrization ($\mu$P) (Yang et al., 2022). Figure 14 shows the assigned learning rates by maximal update parametrization and layer-wise learning rate initialization. Maximal update parametrization scales the batchnorm scale, bias layer, the first convolution and last linear layer w.r.t a base model, while the learning rates values assigned by initialization remains invariant to the model width, except for the final few layers.

