# OpenReview forum: "Initializing the Layer-wise Learning Rate"
_ICLR.cc/2024/Conference — Submitted to ICLR 2024_

### Official Review · Reviewer_R9Q4 · 2023-10-26

**Soundness:** 2 fair
**Presentation:** 3 good
**Contribution:** 2 fair
**Rating:** 3
**Confidence:** 4

**Summary:**

This paper proposes a method to set learning rates for each parameter individually in neural networks.
These learning rates are computed from the reciprocal of the gradient magnitude for each parameter at initialisation time.
Experiments on ImageNet and CIFAR-100 show promising results that confirm the hypothesis that learning rate initialisation can speed up training significantly.

**Strengths:**

- (clarity) The method and results are presented clearly.
 - (significance) Speeding up SGD with a simple learning rate initialisation could be a cost-effective alternative to adaptive optimisation algorithms.

**Weaknesses:**

- (clarity) Layer-wise learning rates, which are probably also addressed in the first edition of tricks of the trade (1998), were probably the only way to make deep networks trainable.
   The main advantage of adaptive optimisers has always been that the painfull process of finding learning rates for each layer is no longer necessary.
 - (clarity) It is unclear how important the choice for $T$ in algorithm 1 is.
   In the experiments, $T$ is the number of batches in one epoch, but there are no ablations for different choices of $T$.
 - (originality) This paper fails to mention its relation to Adagrad (Duchi et al., 2011).
   This is especially relevant because Adagrad can also be interpreted as dividing the learning rate by a running average of the norm.
 - (originality) Algorithm 1 looks a lot like running an adaptive optimiser with a learning rate of zero for a number of mini-batches.
   This connection is completely ignored in the current manuscript.
 - (quality) In this work, the hyper-parameters seem to be shared for the different methods.
   For a proper comparison, hyper-parameters should be tuned for each method individually.
 - (significance) The inspection of the assigned learning rates seem to provide more information about the model than about the method.
   I think these could provide more information when compared to the learning rates of adaptive optimisation algorithms.
 - (significance) The experimental setup is too complex to properly evaluate the merits of the proposed method.
   By evaluating on these large models, confounding factors like learning rate schedules become necessary, making it hard to evaluate the generality of the method.
   Furthermore, these large models typically make it impractical to provide error bars and establish the statistical significance of the presented results.

### Minor Comments
 - There are quite a number of typos (e.g.: touse in abstract, $n$ on line 4 of algorithm 1)
 - Technically, the learning curves of the layer-wise learning rates should be shifted by one epoch, since they are one epoch ahead of the single learning rate baselines.

### References

 - Duchi, J., Hazan, E., & Singer, Y. (2011).
   Adaptive subgradient methods for online learning and stochastic optimization.
   Journal of machine learning research, 12(7). https://www.jmlr.org/papers/v12/duchi11a.html

**Questions:**

1. Please, rewrite the motivation to better reflect the historical evolution of adaptive optimisation methods.
 2. How does this method relate to Adagrad and other adaptive optimization methods?
 3. Is it possible to include simple experiments (cf. Kingma et al., 2015) with error bars?
 4. How much does the learning rate schedule affect the performance of the proposed method?
 5. Is it possible to include a run where Adagrad (or other adaptive methods) iterates the data for one epoch with learning rate zero.
 6. How does the setting above compare to the baseline performance and the proposed layer-wise learning rate?
 7. How important is the choice for $T$ and does this relate in some way to learning rate warmup?
 8. Can you tune the hyper-parameters (most notably the learning rate) for each algorithm individually to provide a fair method comparison?

---

> ### Author Response · Authors · 2023-11-22
> **Response to reviewer R9Q4**
>
> We thank the reviewer for the valuable review and feedback
>
> > Relation to adaptive optimisation methods
>
> We believe that explicit layer-wise learning rates has an effect that is different and mostly independent from adaptive optimization methods. Figure 8 in the Appendix of the updated draft shows the adaptive step size of AdamW $1/(\sqrt{\hat{v}_{t}}+\epsilon)$ when training ResNet-50 for 90 epochs with single and the proposed layer-wise learning rate. The adaptive step sizes show a low to high trend according to layer depth on both single and the proposed method. If the proposed method was an cost-effective alternative to adaptive optimisation algorithms we believe the adaptive step sizes should have flattened out.
>
> We would like to note that the observation layer-wise learning rates can improve training was on experimental settings where adaptive methods are known to achieve lower validation accuracy. We have updated the draft to better reflect the relation with adaptive methods.
>
> > Ablation study on $T$ of Algorithm 1
>
> We performed an ablation study on $T$ when training ResNet-50 for 90 epochs with SGD by measuring the deviation of per-layer assigned learning rates compared to the $T$ used in the paper. We report the results below and have included it in the appendix.
>
> $T$ | Max deviation % | Average deviation % | Validation accuracy %|
> -----------   | :-------------:| :-------------:| :-------------:|
> 1    | 11.02            | 2.66                 | 77.07               |
> 10   | 3.51             | 0.69                 |  77.01              |
> 100  | 1.01             | 0.18                 |   77.04             |
> 1000 | 0.32             | 0.06                 |   77.07             |
> 5004 | 0               | 0                     |    77.15             |
>
> > Additional results with multiple runs
>
> We provide additional results that reinforce the claim that the proposed method has a different effect compared to adaptive optimization methods. We report the result of training ResNet-50 with AdamW for 90 epochs below. While we used a T of 5004, the ablation study of T above showed the choice of T can be as low as 100 and is not significant, and for single learning rate we additionally iterate with learning rate of 0 for 100 iterations. We report the average and standard deviation of three runs below, and it can be seen that layer-wise learning rates have favorable performance even when using adaptive optimization methods.
>
> Initial LR | Single | Layer-wise |
> -----------   |  :-------------:| :-------------:|
> 0.0015 | **75.96±0.11**    | 75.45±0.16 |
> 0.003 |  76.45±0.07 | **76.61±0.15** |
> 0.006 |  75.93±0.10 | **77.00±0.23** |
>
> > Relation with warmup
>
> We consider learning rate warmup to be largely independent and not a setup to be replaced or removed. Our experience is that it is beneficial to both single and layer-wise learning rate, and all experiments are performed with learning rate warmup.
>
> > Regarding the complex experiment setup and confounding factors
>
> While we agree that the experimental setup can be affected by confounding factors such as learning rate schedule and warmup, ImageNet-1k classification has been extensively studied with established baselines and reported hyperparameters. We believe evaluating on such setups is effective in demonstrating methods that work on large datasets and architectures.

---

### Official Review · Reviewer_cEsi · 2023-11-01

**Soundness:** 2 fair
**Presentation:** 2 fair
**Contribution:** 2 fair
**Rating:** 3
**Confidence:** 3

**Summary:**

This paper proposes a method to assign a learning rate to each layer. This layer-wise learning is computed by using the norm of the backpropagated gradients. Basically, the learning rate of assigned to a layer $l$ is inversely proportional to the square root of the $\mathcal{L}^1$-norm of the backpropagated gradient (according to the tensor of weights of $l$).

To evaluate their heuristic, the authors run two series of experiments: one with a single learning rate, and one with their heuristic. The tested setups include two optimizers: SGD and AdamW. In each tested case, the reported performance is greater with their method than with theit single learning rate counterparts.

**Strengths:**

## Originality

To my knowledge, the proposed heuristic for a layer-wise learning rate is new.

## Clarity

Overall, the proposed method is easy to understand.

## Quality

The authors have well explained (introduction of Section 3) why one should look for a working and well-justified heuristic for layer-wise learning rates, computed *before* training. This problem deserves to be studied, both from a practical and a theoretical point of view.

The experimental results are very encouraging.

**Weaknesses:**

## Clarity

The idea behind Algorithm 1 is easy to understand, but several details are missing or seem to be erroneous:
 * line 4: replace $n$ by $t$;
 * apparently, the $G_i$ are incremented $T$ times, but they are not normalized by $T$ or any other quantity depending on $T$. So, two questions arise: how do we choose $T$? Or should we normalize the $G_i$ somewhere?

More importantly, many choices in Algorithm 1 are not explained by the authors:
 * line 8: why do the authors use the $\mathcal{L}^1$-norm over $\mathbf{g}$, and not the $\mathcal{L}^2$-norm or any other norm?
 * line 10: why choosing the inverse of the square root of $G_i$, and to the inverse of $G_i$ or any other quantity?
 * lines 11-13: what is the justification for such a computation?

Overall, Algorithm 1, which describes the entire method proposed by the authors, is incomplete and lacks justification. This crucial weakness can be solved by adding subsections in Section 1, proving mathematically all the choices made in Algorithm 1 (at least in simple cases). Otherwise, these choices remain arbitrary.

**Questions:**

Could the authors provide at least a short analysis of their method in simple cases, or in extreme cases (layer size tending to infinity)? It would be interesting to observe what happens at the first training step.

What do the authors think about the paper *Neural tangent kernel: Convergence and generalization in neural networks*, Jacot et al., 2018? In this paper, each weight tensor is scaled by $1/\sqrt{f_{\text{in}}}$. This setting, combined with a unique learning rate for all layers, is equivalent to the "normal" setting (without scaling) with a learning rate per layer, proportional to $1/f_{\text{in}}$. How the learning rates computed by the authors compare to these?

Experimental results: are the results consistent when we change the learning rate? Does the proposed method perform better than the "single lr method" in any circumstance?

---

> ### Author Response · Authors · 2023-11-22
> **Response to reviewer cEsi**
>
> We thank the reviewer for the valuable review and feedback
>
> > Details on Algorithm 1
>
> For $T$ we find a value of 100 or higher is sufficient to obtain resonable values. $G_{i}$ is incremented $T$ times in line 8, and is used to calculate the relative learning rates in line 10. After obtaining the relative values, the normalization is performed on lines 11-13 such that the average per-parameter learning rate is one.
>
> > Choice of $\mathcal{L}^1$ over $\mathbf{g}$ and inverse of the square root of $G_i$
>
> Algorithm 1 is developed with the interpretation that the gradient magnitude itself is a objective to be minimized, so Algorithm 1 uses the gradient magnitude directly instead of performing $\mathcal{L}^2$-norm. That said, we find using $\mathcal{L}^2$-norm over the per-parameter gradient instead of $\mathcal{L}^1$-norm assigns very similar layer-wise learning rates, with an average deviation of 0.97%, and achieves similar performance of 77.24% on ResNet-50 when trained with SGD for 90 epochs.
>
> The square root over $G_i$ in line 10 is performed because without it we found the difference in scale of assigned learning rates varies widely and results in drastically reduced performance of 75.99%.
>
> > Comparison to $1/f_{\text{in}}$ learning rate scaling
>
> On ResNet-50, we find $1/f_{\text{in}}$ tends to assign assigns higher learning rates to initial layers and lower learning rates to later layers, which is directly opposite to the proposed method. Under the same unit per-parameter normalization of line 11-13 in Algorithm 1, $1/f_{\text{in}}$ scaling assigns 6.295 to the first 7*7 convolution layer, and 14.459 to the directly succeding convolution layer, which is in stark contrast to the 0.038 and 0.109 assigned by Algorithm 1 in Figure 4.
>
> For ViT-S/16, $1/f_{\text{in}}$ scaling assigns identical learning rates to all the query, key and value layers in the self attention block, while the proposed method assigns higher learning rates to the query and key layers. We believe the practice of normalizing query and key layers in self attention [1][2] further validates the empirical significance of our method.
>
> [1] Henry, Alex, et al. "Query-Key Normalization for Transformers." Findings of the Association for Computational Linguistics: EMNLP 2020. 2020.
>
> [2] Dehghani, Mostafa, et al. "Scaling vision transformers to 22 billion parameters." International Conference on Machine Learning. PMLR, 2023.
>
> > Results under different learning rates
>
> We provide additional results when training ResNet-50 with half and double the base learning rate for 90 epochs with SGD below. It shows that the proposed method performs well over various learning rates.
>
> Initial LR | Single | Layer-wise |
> -----------   |  :-------------:| :-------------:|
> 0.05 | 76.61    | **76.67** |
> 0.1 | 76.80    | **77.12** |
> 0.2 | 76.33    | **76.90** |

---

> > ### Comment · Reviewer_cEsi · 2023-11-22
> >
> > My main concern about this paper remains the absence of a section where the choices made by the authors in Algorithm 1 are grounded by some theoretical evidence (with a theorem or a heuristic, even in a very simple case).
> >
> > My argument is: it is always possible to invent some method adding several hyperparameters, fine-tune these parameters by an expensive try-and-error process in a small number of configurations, and beat some baseline in these configurations. It is neither surprising nor useful to beat the baseline (by a narrow margin) by adding hand-tuned hyperparameters. Since the benefits of the proposed algorithm are not outstanding in the proposed setups, at least a theoretical explanation is needed.
> >
> > > Algorithm 1 uses the gradient magnitude directly instead of performing $\mathcal{L}^2$-norm
> >
> > Apparently, there is a lack of clarity when using the term "gradient magnitude". In many of the papers cited (Zhang 2020b, Yu 2017, Balles 2018, etc.), "gradient magnitude" either refers explicitly to the $\mathcal{L}^2$-norm, either means informally "the size of the gradient", without being specific. In any case, "gradient magnitude" does not mean unambiguously "$\mathcal{L}^1$-norm"

---

### Official Review · Reviewer_LTJF · 2023-11-07

**Soundness:** 2 fair
**Presentation:** 2 fair
**Contribution:** 2 fair
**Rating:** 3
**Confidence:** 3

**Summary:**

This paper proposes a systematic layer-wise learning rate adjusting scheme according to the layer-wise gradient magnitude at initialization, improving training performance and stability on convolutional and transformer architectures. Competitive results on convolutional and transformer architectures on CIFAR100 and ImageNet-1k validate the proposed hypothesis.

**Strengths:**

The method is easy to understand.
The experiment results are convincing, removing the fluctuations and improving accuracy.

**Weaknesses:**

1. Lots of typos and confusing statement. Such as "Figure 7: Train loss for 2-layer MLP trained on CIFAR-10 trained.", Sec.2: "Another direction previous works indirectly modified the learning rate is through the use of scale factors..."
2. In Algorithm 1, the reason of choice of T and corresponding ablation study is missing which may be vital to the performance of the proposed algorithm.
3. The novelty is limited; more theoretical analyses are needed.
4. Related works are not clear enough.

**Questions:**

1. What is the motivation of proposed Algorithm 1?
2. Why layer-wise learning rate scheme performs not so good on Swin-T and ConvNeXt-T when using AdamW? ResNet-50 and SGD are no longer mainstream models or algorithms in 2023. What are the impacts of proposed algorithms on different model structures and modules?
3. Figure 6 is not intuitive, even seems that single way is better than proposed methods.
4. More ablation studies are needed to validate the influence of different hyper-parameters in Algorithm 1.

---

> ### Author Response · Authors · 2023-11-22
> **Response to reviewer LTJF**
>
> We thank the reviewer for the valuable review and feedback
>
> > Motivation of proposed Algorithm 1
>
> Algorithm 1 adjusts the layer-wise learning rate to regularize architecture-induced convergence bias, measured as the gradient magnitude at initialization. It is motivated from an observation that layer-wise learning rate can further improve training of SGD in settings where adaptive methods are considered to have lower validation accuracy due to higher generalization gap.
>
> > Effect and performance of different architectures
>
> Inspecting the assigned learning rates show that the convergence bias of Swin-T and ConvNeXt-T resembles that of vision transformer more than convolutional networks. While there could be various factors, we think lack of hyperparameter tuning could be a major reason for the lack of performance gain on Swin-T and ConvNeXt-T when using AdamW. Given how sensitive performance is to learning rates, we believe achieving competitive performance when learning rates of layers can differ by an order of magnitude is still a very interesting phenomenon.
>
> Swin-T and ConvNeXt-T are complex architectures that incorporate design choices of both convolutional and transformer architectures. Swin-T incorporates hierarchical feature maps which is similar to feature map hierarchy in convolutional networks, and ConvNeXt-T stems from introducing vision transformer designs on convolutional networks. We mainly focus on ResNet and vision transformer as they are representative of the two main architecture family in vision tasks, and we believe performance on SGD is a good indication of training stablity and generalizability.
>
> > In Figure 6, single seems better than proposed method
>
> In terms of final accuracy we agree Figure 6 shows using a single learning rate is better in such settings. However CIFAR-100 differs widely from ImageNet-1k in terms of dataset size, resolution and difficulty, and we believe methods that are beneficial on larger dataset size and difficulty is of interest even if it isn't necessarily beneficial on smaller tasks.
>
> > Ablation studies to validate the influence of hyper-parameters in Algorithm 1
>
> We performed an ablation study on $T$ when training ResNet-50 for 90 epochs with SGD by measuring the deviation of per-layer assigned learning rates compared to the $T$ used in the paper. We report the results below and have included it in the appendix.
> $T$ | Max deviation % | Average deviation % | Validation accuracy %|
> -----------   | :-------------:| :-------------:| :-------------:|
> 1    | 11.02            | 2.66                 | 77.07               |
> 10   | 3.51             | 0.69                 |  77.01              |
> 100  | 1.01             | 0.18                 |   77.04             |
> 1000 | 0.32             | 0.06                 |   77.07             |
> 5004 | 0               | 0                     |    77.15             |

---

### Official Review · Reviewer_5Czw · 2023-11-10

**Soundness:** 3 good
**Presentation:** 2 fair
**Contribution:** 3 good
**Rating:** 6
**Confidence:** 3

**Summary:**

This paper focused on the problem about initialization method of layer-wise learning rate. The authors use gradient magnitude as a
measure of architecture-induced convergence bias. Based on that, they try to adjust the layer-wise learning rate opposite to its gradient magnitude at initialization. The experimental results illustrate that the proposed initialization method can obtain a better performance on CIFAR, ImageNet.

**Strengths:**

1. This paper focus on an important problem. In my past experience about neurwal network training, layer-wise learning rate is very sensitive to the initialization method of each layer. For example, LAMB use the ratio between weight norm and gradient norm to determine the layer-wise learning rate.
2. The proposed method is very easy to understand. We can estimate gradient magnitude and then determine the layer-wise learning rate.

**Weaknesses:**

1. I'm not sure whether the proposed method can scale. Although the proposed method and intuition are easy to understand, the method is still not simple enough. So that make me consider the performance when we scale to a very large model, such as a language model, and whether this can be a general method. I know this is very difficult and I'm just considering. If possible, you could provide some results on NLP task.
2.  You need to compare the proposed method with more layer-wise optimization method, such as LARS and LAMB. I noticed that the main baseline is SGD and Adam. and these methods are not layer-wose methods. Although their performance is very strong, I think LRAS / LAMB can also further improve the performance of SGD / Adam. To better illustrate the performance gain of your method, maybe you should provide these results on layer-wise method.

**Questions:**

1. I would like to ask the training cost of Algorithm 1. Since the method need to estimate the gradient magnitude and other methods don't need it. Therefore, I would like to ask the cost, such as time. In addition, the proposed need to use T steps in Algorithm 1 and that means we need more steps to finish the training with the proposed method. If we add these T steps to the baselines, such as SGD and Adam, whether we can further improve their performance?

---

> ### Author Response · Authors · 2023-11-22
> **Response to reviewer 5Czw**
>
> We thank the reviewer for the valuable review and feedback
>
> > Regarding the simplicity of the method and comparison with LARS, LAMB
>
> We would like to emphasize that while LARS, LAMB are also layer-wise methods, the underlyring mechanisms and interpretation are very different in that LARS and LAMB relies on per-layer gradient normalization and additional weight norm scaling, while the proposed method is concered with architecture-induced convergence bias. In practice it means the learning rates of LARS and LAMB is insensitive to gradient of other layers, and due to the additional weight norm scaling layers initialized 0 are in principle not updated.
>
> We have provided results of layer-wise gradient normalization techniques on ResNet-50 with SGD on Table 2. We found that the batchnorm scale and bias layers had to be excluded from LAMB scaling for LAMB to achieve competitive performace, perhaps due to its additional weight norm scaling. In that sense, we believe the proposed method can be considered simpler as the step size remains dependent only the gradient and does not modifiy the optimizer.
>
> > Training cost of Algorithm 1
>
> We performed an ablation study on $T$ when training ResNet-50 for 90 epochs with SGD by measuring the deviation of per-layer assigned learning rates compared to the $T$ used in the paper. We report the results below and have included it in the appendix. We also ran a baseline experiement of 91 epochs, but found the final validation accuracy to be lower that the reported 76.80%, suggesting that other factors are more dominant compared to a single epoch difference.
> $T$ | Max deviation % | Average deviation % | Validation accuracy %|
> -----------   | :-------------:| :-------------:| :-------------:|
> 1    | 11.02            | 2.66                 | 77.07               |
> 10   | 3.51             | 0.69                 |  77.01              |
> 100  | 1.01             | 0.18                 |   77.04             |
> 1000 | 0.32             | 0.06                 |   77.07             |
> 5004 | 0               | 0                     |    77.15             |

---

### Meta-Review · Area_Chair_WFpc · 2023-12-08

**Metareview:**

- This submission proposes a method to tune the init learning rates layer-wisely for neural networks based on the magnitude of the grad $\ell$_1 norm.
- The experimental results show that the proposed methods could improve the generalization ability of the existing optimizers.
- However, the existing submission still has several flaws: 1) the motivation is unclear; 2) the connection to the adaptive methods especially to the layer-wise adaptive methods should be discussed in detail (only showing the numerical results is not enough); 3) the current experimental setting may be out-of-date, for example, for Swin-T and ViT-S, the acc@1 is over 80% in the common advanced setting.

Based on the flaws, this version is not suitable for publication now.

**Justification For Why Not Higher Score:**

- The submission contains several flaws and may not be solved easily.

**Justification For Why Not Lower Score:**

N/A

---

### Decision · Program_Chairs · 2024-01-16

Reject